# Efficacy and safety of cisplatin + docetaxel + 5-FU + leucovorin + methotrexate and epirubicin combination chemotherapy for advanced esophageal cancer

Sung-Chi Yu[1], Shao-Syuan Tong[1], Yi-Ling Chen[1], Ya-Fu Cheng[1], Jin-Ching Lin[2], Ching-Yuan Cheng[1,3], Chang-Lun Huang[1], Wei-Heng Hu[1], Bing-Yen Wang[1,3]*

**1** Division of Thoracic Surgery, Department of Surgery, Changhua Christian Hospital, Changhua, Taiwan, **2** Department of Radiation Oncology, Changhua Christian Hospital, Changhua, Taiwan, ROC, **3** Department of Post-Baccalaureate Medicine, College of Medicine, National Chung Hsing University, Taichung, Taiwan

\* 156283@cch.org.tw

## Abstraction

Esophageal cancer is a devastating disease. The cisplatin and 5-FU regimen is the most widely used concurrent chemoradiotherapy protocol for metastatic, unresectable advanced esophageal cancer in Asia. However, its effectiveness remains limited due to unsatisfactory outcomes. Therefore, we conducted a retrospective study to evaluate the efficacy and toxicity of Cisplatin + 5-FU combined with docetaxel, leucovorin, methotrexate, and epirubicin (CDFLME), a first-line multi-agent chemotherapy regimen used for the treatment of advanced esophageal cancer in Taiwan. We enrolled 94 patients in our study from January 2018 to June 2022. All patients were diagnosed with metastatic or unresectable advanced esophageal cancer. Among them, 81 patients received fluorouracil + cisplatin regimen serving as the control group, while 13 patients received the CDFLME combination regimen. Significant improvements were observed in the CDFLME group compared to the fluorouracil + cisplatin group in overall survival time, complete response rate, and disease control rate. No significant differences were noted in treatment-related deaths or grade 3–4 adverse events, except for grade 1–2 mucositis.

Based on these findings, we conclude that the CDFLME regimen is a promising alternative treatment with relatively minor adverse events and is an effective protocol for patients with advanced esophageal cancer.

## Introduction

Esophageal cancer is a devastating disease, ranking as the ninth leading cause of cancer-related deaths worldwide [1,2]. While adenocarcinoma incidence is increasing in Western nations, squamous cell carcinoma remains the predominant pathology in Asia [3]. Characterized by early lymphatic and hematogenous spread, nearly half of esophageal cancer cases present with metastatic disease at diagnosis, resulting in a poor prognosis. The 5-year overall survival rates for esophageal cancer patients range between 15% and 25% [4].

**Data availability statement:** The data cannot be shared publicly due to restrictions imposed by the Institutional Review Board of Changhua Christian Hospital. However, data are available from the Changhua Christian Hospital Big Data Center for researchers who meet the criteria for access to confidential data. Requests for data access can be made through the Big Data Center website: https://dpt.cch.org.tw/layout/layout_7/item.aspx?cID=263&ID=7341, or by contacting the center via email at cchbd7341@gmail.com. The data underlying the results presented in this study are available upon reasonable request and with permission from the Changhua Christian Hospital Big Data Center.

**Funding:** The author(s) received no specific funding for this work.

**Competing interests:** The authors have declared that no competing interests exist.

The treatment options for esophageal squamous cell carcinoma (ESCC) include surgery, radiation, and chemotherapy [5–8]. Most chemoradiotherapy regimens utilize doublets of cisplatin combined with fluorouracil, as these are two of the most active single agents against squamous carcinomas. The efficacy of this combination has been well-documented since the 1990s [9]. More than half of patients are ineligible for surgery or present with metastatic disease at diagnosis. Concurrent chemoradiotherapy has become a standard approach for managing unresectable advanced ESCC. The RTOG trial 85−01 [10] established the efficacy of cisplatin, 5-FU, and concurrent radiotherapy (50.4Gy) for patients with T1-3N0-1M0 esophageal cancer.

This regimen is the most widely used concurrent chemoradiotherapy protocol for metastatic, unresectable advanced esophageal cancer in Asia. Reported response rates range from 25% to 45%, with a median overall survival of 6–10 months in phase II and III trials [11–15]. However, the outcomes remain unsatisfactory for advanced esophageal cancer patients, highlighting the need for more effective therapeutic options.

Recent investigations have explored the feasibility and safety of definitive chemoradiotherapy in patients with synchronous head and neck squamous cell carcinoma and esophageal cancer. These studies demonstrated low mortality rates and acceptable morbidity [16]. Notably, head and neck anticancer agents have shown promise in the treatment of esophageal cancer. Despite this, few studies have assessed the feasibility and safety of head and neck cancer chemotherapy regimens in esophageal cancer patients.

In 2016, J.-C. Lin [17] designed a novel weekly Cisplatin + 5-FU combined with docetaxel, leucovorin, methotrexate, and epirubicin (CDFLME) induction chemotherapy regimen, which demonstrated a higher response rate and a lower incidence of Grade 3/4 mucositis and neutropenia compared to the outcomes reported for TPF/PF regimens in the literature for locally advanced Head and neck squamous cell carcinoma (HNSCC).

Currently, there are limited studies evaluating the efficacy and safety of head and neck cancer chemotherapeutic regimens in advanced esophageal cancer patients [17,18]. Therefore, our study aimed to evaluate the efficacy and toxicity of a first-line chemotherapy regimen CDFLME in advanced esophageal cancer patients in Taiwan.

## Materials and methods

### Database and study sample

This study was conducted as a retrospective study and is registered with IRB number 231101. The date of approval was November 21, 2023. For studies involving human participants, patient consent was waived. The data used for research purposes were accessed on November 21, 2023. In this retrospective study, the requirement for informed consent was waived by the Institutional Review Board (IRB) of Changhua Christian Hospital (IRB number: 231101)).

All patients were enrolled from January 2018 to June 2022 across institutions associated with Changhua Christian Hospital.

The data cannot be shared publicly due to restrictions imposed by the Institutional Review Board of Changhua Christian Hospital. However, data are available from the Changhua Christian Hospital Big Data Center for researchers who meet the criteria for access to confidential data. Requests for data access can be made through the Big Data Center website: https://dpt.cch.org.tw/layout/layout_7/item.aspx?cID=263&ID=7341, or by contacting the center via email at cchbd7341@gmail.com. The data underlying the results presented in this study are available upon reasonable request and with permission from the Changhua Christian Hospital Big Data Center.

All monitoring visits were completed prior to the commencement of this retrospective study. The data used in this study were obtained from the medical records of Changhua Christian Hospital, ensuring accuracy and completeness.

Eligibility criteria included a Pathologically confirmed diagnosis of metastatic or unresectable advanced esophageal cancer (adenocarcinoma or squamous cell carcinoma) through tissue biopsy. All patients with esophageal cancer underwent diagnostic procedures including CT, PET/CT, upper gastrointestinal panendoscopy, endoscopic ultrasound, and abdominal ultrasound. For this retrospective study, all advanced esophageal cancer patients underwent concurrent chemoradiotherapy. Subsequently, these patients were stratified into two chemotherapy regimens: the cisplatin + fluorouracil (PF) group and the CDFLME group.

Inclusion criteria included a performance status of ≤2 as determined by the Eastern Cooperative Oncology Group (ECOG) and the presence of at least one measurable lesion as determined by the Response Evaluation Criteria in Solid Tumors (RECIST), version 1.1 [19]. Progressive disease is defined as at least a 20% increase in the sum of diameters of up to 5 target lesions (2 lesions/organ), taking as reference the smallest sum on studywith an absolute lesion increase of at least 5 mm or the appearance of new lesions. Patients who meet the criteria for surgery during follow-up and subsequently undergo the procedure will be excluded. A complete response is defined as the disappearance of all target lesions, and a partial response is defined as at least a 30% decrease in the sum of diameters of target lesions, taking as reference the baseline sum diameters. Stable disease is defined as meeting the criteria for neither progressive disease nor for a partial response. The overall response rate is defined as the percentage of patients who have a partial or complete response to therapy. Disease Control Rate is defined as the proportion of patients who respond to treatment, including those with complete response, partial response, or stable disease, according to RECIST criteria [19].

Since there is no consensus on a temporal definition of Treatment-related death, we adopted the stricter criteria of Adelstein et al [20]. Defining Treatment-related death as death occurring from CRT start until a month from the end of CRT.

A total of 94 patients were enrolled in this study. Eighty one patients who received the PF regimen were placed into the PF group, and thirteen patients who received the CDFLME regimen were placed into the CDFLME group. Patients were also required to have sufficient laboratory data, which included hematologic, renal, and hepatic function, absolute neutrophil count, and total bilirubin, aspartate transaminase, and alanine transaminase levels. Additionally, clinical baseline data were obtained from the Changhua Christian Hospital database, including age, sex, esophageal tumor location, clinical stage, and ECOG performance status. Exclusion criteria comprised patients lost to follow up and those who refused chemotherapy or systemic treatment. This study was approved by the institutional review board of the hospital, and all patients provided written informed consent before enrollment. Tumor grading was based on the World Health Organization's classification system, and staging was performed according to the 7th edition of the tumor, node, metastasis (TNM) staging system. Our research protocol meets the guidelines of the Taiwan Society of Thoracic Surgeons' requirements for a specialist medical center.

### Treatment schedule and dosage modification

Patients in the PF group were treated according to the following schedule: cisplatin (intravenously, 75–100 mg/m²) was administered on day 1, followed by a continuous infusion of fluorouracil (500–1000 mg/m²) over 24 hours daily from days 1–4. This cycle was repeated every 28 days for 2–4 cycles, consisting of 2 cycles with radiation followed by 2 cycles

without radiation [3]. Patients in the CDFLME group were treated according to the following schedule: (1) cisplatin 60 mg/m2,day 1, (2) docetaxel 50 mg/m2, day 8, (3) 5-fluorouracil 2500 mg/m2 + leucovorin 250 mg/m2, day 15, (4) epirubicin 30 mg/m2 + methotrexate 30 mg/m2, day 22, were alternatively delivered once per week, with one cycle every 4 weeks, for a total of 3–4 cycles. [17](Fig 1). Dosage adjustments were made based on observed adverse events. The definitive radiotherapy dosage ranged from 50 to 50.4 Gy (1.8–2 Gy/day). Dose escalation to the gross tumor volume (60–66 Gy) was considered feasible using the intensity modulated radiation therapy technique.

Throughout irradiation, patients were monitored at least once a week for status checks, including vital signs, weight, and blood counts. Prophylactic antiemetics were administered as needed. Additionally, antacids and antidiarrheal medications were prescribed when indicated. Oral and/or enteral nutrition was considered if estimated caloric intake was less than 1500 kcal/day. Adequate enteral and/or intravenous hydration was maintained throughout the definitive concurrent chemoradiation therapy. Conservative management of chemical mucositis focuses on symptom relief, oral care, nutritional support, and maintaining moisture. Topical anesthetics and analgesics can help manage pain, while regular rinsing with saline or baking soda solutions ensures oral hygiene.

The subsequent therapy is depends on prior therapy response and performance status. In addition, we will implement the same treatment approach, including palliative and best supportive care, for both groups of patients. For patients meeting the criteria for immunotherapy, immune checkpoint inhibitors were administered regardless of whether they received neoadjuvant chemoradiotherapy.

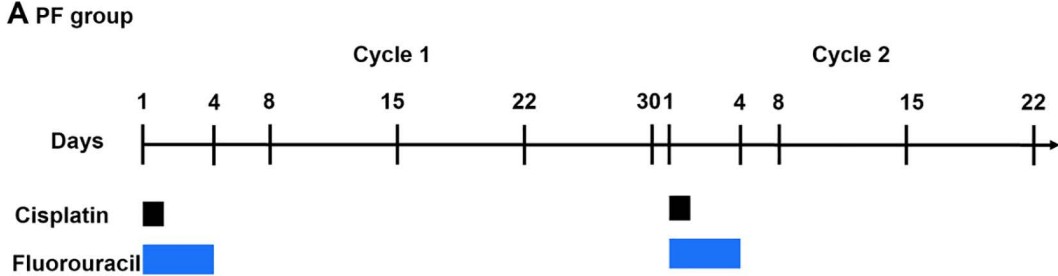

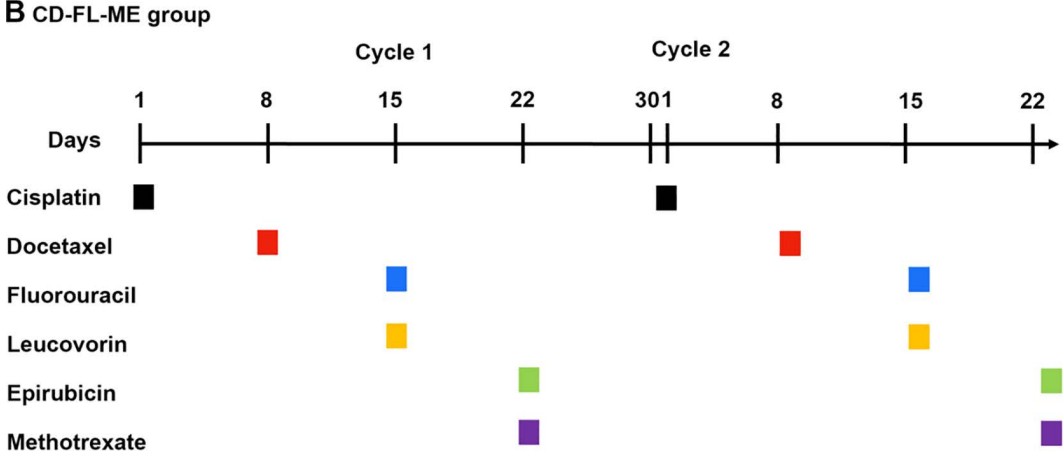

**Fig 1. The schedule of chemotherapy.**

## Assessment of tumor response and toxicity

A baseline CT scan or MRI scan was performed when patients underwent tumor staging, and subsequent scans were repeated every 3 months to assess tumor response. Evaluation of tumor response was conducted in accordance with version 1.1 of the RECIST [19]. When a tumor exhibits distant metastasis to sites including the lungs, liver, bones, distal lymph nodes, adrenal gland or brain, it is considered disease progression. Toxicity assessments were carried out prior to each cycle of chemotherapy using version 5.0 of the National Cancer Institute Common Terminology Criteria for Adverse Events.

## Statistical analysis

The primary endpoint of this study was the response rate. Secondary endpoints included the 2-year overall survival rate and chemotherapy toxic effects. Survival time was defined as the period from the confirmation of malignancy to either death or until June 2022. Overall survival rates were estimated using the Kaplan-Meier method, and the log-rank test was employed to compare the PF group and the CDFLME group. Differences in survival estimates were calculated using the Cox proportional hazards regression model, which was stratified for hospital volume and adjusted for known prognostic factors. All statistical calculations were performed using the Statistical Analysis System, version 9.3, and Statistical Product and Service Solutions, version 20. Multivariate analyses were also conducted using the Cox proportional hazards model. All variables in the univariate analysis were entered into the multivariate analysis, where a p-value of less than 0.05 was considered statistically significant.

## Results and discussion.

### Clinical baseline characteristics

From January 2018 to June 2022, a total of 94 patients from institutions associated with Changhua Christian Hospital were enrolled in this study. Baseline characteristics of the patients are summarized in Table 1. We divided the patients into the PF group and the CDFLME group. There were 81 patients who received fluorouracil+cisplatin regimens and 13 patients who received cisplatin+docetaxel+5-FU+leucovorin+methotrexate+epirubicin combination regimens. Most patients (90 [95.7%]) were male. The median age was 60 years (range 53.0–66.3 years). A minority of patients had metastatic disease at baseline in both the PF group (42.0%) and the CDFLME group (15.4%). In the PF group, 78 cases had an ECOG performance status of 0–2, and 3 cases had an ECOG performance status of 3–4. In the CDFLME group, all 13 cases had an ECOG performance status of 0–2. Fourteen patients died in the PF group by the time of the first follow-up (3 months), and 1 patient died in the CDFLME group by the time of the first follow-up.

### Clinical efficacy

We assessed the efficacy within the PF group and the CDFLME group (Table 2). In the PF group, only 1 patient achieved a complete response and 25 patients achieved a partial response. The response rate of the PF group was 32.1%, and the disease control rate was 53.1%. In the CDFLME group, 3 patients achieved a complete response and 4 patients achieved a partial response. The response rate of the CDFLME group was 53.8%, and the disease control rate was 84.6%. There were significant differences in complete response and disease control rate between the two groups. However, there were no significant differences in partial response rate, stable disease rate, and overall response rate between the two groups

### Toxicity and side effects

Toxicity was assessed in all 94 patients since each patient received at least one dose of chemotherapy. Treatment-related adverse events (TRAEs) are summarized in Table 3. TRAEs occurred in 47 out of 81 patients (58.0%) in the PF group and

**Table 1. Clinical baseline characteristics of patients.**

| Variable | Patients | PF | CDFLME | p-value [a] |
|---|---|---|---|---|
| Total | 94 | 81 | 13 | |
| Age | 60.00 (53.00-66.25) | 60.00 (53.50-67.00) | 56.00 (51.50-62.50) | 0.270 |
| Sex | | | | |
| Male | 90(95.7%) | 78(96.3%) | 12(92.3%) | 0.454 |
| Female | 4(4.3%) | 3(3.7%) | 1(7.7%) | |
| Location | | | | |
| Ce | 3(3.2%) | 2(2.5%) | 1(7.7%) | 0.621 |
| Ut | 23(24.5%) | 20(24.7%) | 3(23.1%) | |
| Mt | 32(34.0%) | 27(33.3%) | 5(38.5%) | |
| Lt | 36(38.3%) | 32(39.5%) | 4(30.8%) | |
| TNM Clinical stage | | | | |
| T | | | | |
| T1/T2 | 20(21.3%) | 14(17.3%) | 6(46.2%) | 0.029* |
| T3/T4 | 74(78.7%) | 67(82.7%) | 7(53.8%) | |
| N | | | | |
| N- | 11(11.7%) | 8(9.9%) | 3(23.1%) | 0.176 |
| N+ | 83(88.3%) | 73(90.1%) | 10(76.9%) | |
| M | | | | |
| M0 | 58(61.7%) | 47(58.0%) | 11(84.6%) | |
| M1 | 36(38.3%) | 34(42.0%) | 2(15.4%) | 0.122 |
| Stage | | | | |
| I | 5(5.4%) | 1(1.2%) | 4(30.8%) | 0.001* |
| II | 10(10.6%) | 8(9.9%) | 2(15.4%) | |
| III | 22(23.4%) | 18(22.2%) | 4(30.8%) | |
| IVA | 21(22.3%) | 20(24.7%) | 1(7.6%) | |
| IVB | 36(38.3%) | 34(42.0%) | 2(15.4%) | |
| ECOG status | | | | |
| 0–2 | 91(96.8%) | 78(96.3%) | 13(100.0%) | >0.999 |
| 3–4 | 3(3.2%) | 3(3.7%) | 0(0.0%) | |

[a]:Mann-Whitney U test or chi-square test

PF: cisplatin+5-FU

CDFLME: cisplatin+5-FU with docetaxel, leucovorin, methotrexate, and epirubicin

Ce: cervical esophagus; Ut: upper thoracic esophagus; Mt: middle thoracic esophagus;

Lt: lower thoracic esophagus

N-: refers to clinical stage of N0 of cancer

N+: refers to clinical stages N1, N2, and N3 of cancer

ECOG: Eastern Cooperative Oncology Group Performance Status Scale

in 8 out of 13 patients (61.5%) in the CDFLME group. Grade 3–4 TRAEs were reported in 30 out of 81 patients (37.0%) in the PF group and in 5 out of 13 patients (38.5%) in the CDFLME group.

The most common grade 3–4 hematologic toxicity was neutropenia. The most common grade 3–4 non-hematologic toxicity was poor appetite. There were no significant differences between the two groups in any grade 1–2 TRAE or grade 3–4 TRAE, except for mucositis. Grade 1–2 mucositis occurred in 2 patients in the CDFLME group and 1 patient in the PF group (p=0.046).

**Table 2. Clinical efficacy in two groups.**

| | PF | CDFLME | p-value |
|---|---|---|---|
| | (81) | (13) | |
| Complete response | 1(1.2%) | 3(23.1%) | 0.008* |
| Partial response | 25(30.9%) | 4(30.8%) | >0.999 |
| Stable disease | 17(21.0%) | 4(30.8%) | 0.478 |
| Progressive disease | 24(29.6%) | 1(7.7%) | 0.173 |
| Treatment related death | 14(17.3%) | 1(7.7%) | 0.685 |
| Overall response rate | 26(32.1%) | 7(53.8%) | 0.209 |
| Disease control rate | 43(53.1%) | 11(84.6%) | 0.033* |

Treatment-related death: death occurring from CRT start until a month from the end of CRT.

Overall response rate: includes partial responses and complete responses.

Disease control rate: includes partial responses, complete responses and stable disease observations

**Table 3. Definite chemotherapy toxicity between two groups.**

| | PF | | CDFLME | | p-value |
|---|---|---|---|---|---|
| No. of patients with any TRAE | 47(58.0%) | | 8(61.5%) | | 0.811 |
| No. of patients with grade 3–4 TRAEs among all patient with TRAEs | 30(63.8%) | | 5(62.5%) | | >0.999 |
| | Grade 3–4 | | | Grade 1–2 | |
| | CDFLME(5) | PF(30) | p-value | CDFLME(3) | PF(17) | p-value |
| Nausea or vomiting | 0 | 3 | >0.999 | 1 | 9 | >0.999 |
| Late diarrhea | 0 | 1 | >0.999 | 0 | 0 | |
| Mucositis | 2 | 5 | 0.256 | 2 | 1 | 0.046* |
| Anemia | 0 | 6 | 0.561 | 0 | 0 | |
| Neutropenia | 4 | 20 | >0.999 | 0 | 0 | |
| Thrombocytopenia | 1 | 3 | 0.139 | 0 | 0 | |
| Poor appetite | 1 | 8 | >0.999 | 0 | 11 | 0.074 |
| Acute kidney injury | 0 | 2 | >0.999 | 0 | 0 | |

TRAE: treatment-related adverse event

Treatment-related deaths occurred in 14 out of 81 patients in the PF group (17.3%, primarily due to neutropenia leading to septic shock) and in 1 out of 13 patients in the CDFLME group (7.7%, primarily due to neutropenia, fever, and pneumonia).

A total of 5 out of 13 patients in the CDFLME group expired during subsequent follow up. one of whom died due to treatment-related death. The specific causes of death and treatments are discussed in Table 4

## Overall survival

The 1-year overall survival rate was 20.9% in the PF group compared to 69.2% in the CDFLME group. The 2-year overall survival rate was 15.7% in the PF group compared to 61.5% in the CDFLME group. Additionally, the mean overall survival time was 10.69 months in the PF group and 19.18 months in the CDFLME group; all of these differences were statistically significant (Table 5).

The 2-year survival rate and mean survival time were assessed and stratified based on each clinical characteristic, including age, sex, tumor location, TNM clinical stage, treatment-related adverse event (TRAE) grade, ECOG

**Table 4. All causes of mortality and treatments in the CDFLME group.**

| Case | Age/Sex | Clinical stage | ECOG | Clinical efficacy | Survival time(months) | Treatment-related adverse events | Description/Treatment |
|------|---------|----------------|------|-------------------|----------------------|----------------------------------|----------------------|
| Case 1 | 57/Male | cT3N2M0 | 0 | Stable disease | 11.5 | nil | Cachexia/support care |
| Case 2 | 67/Male | cT1N0M0 | 1 | Stable disease | 9.9 | nil | Pneumonia, tumor bleeding/antibiotics, support care |
| Case 3 | 63/Male | cT4N3M0 | 1 | Progress disease | 12.5 | Grade 3–4 neutropenia and Thrombocytopenia | Febrile Neutropenia/antibiotic, G-CSFs |
| Case 4 | 51/Male | cT3N3M0 | 1 | Partial response | 20.6 | Grade 1–2 nausea/vomiting Grade 3–4 neutropenia | Pneumonia/antibiotic |
| Case 5 | 50/Male | cT1N0M0 | 2 | Expired before follow up | 2.7 | Grade 3–4 neutropenia and Thrombocytopenia | Febrile Neutropenia/antibiotic, G-CSFs |

G-CSFs: granulocyte colony-stimulating factors

**Table 5. Mean OS time and 1-year and 2-year OS rates.**

| | PF (n=81) | CDFLME (n=13) | p-value |
|---|-----------|---------------|---------|
| Mean OS, months | 10.69 (9.12-12.25) | 19.18 (15.34-23.01) | <0.001 |
| 1-year OS rate | 20.9%(11.9%−29.9%) | 69.2%(43.6%−94.8%) | 0.007 |
| 2-year OS rate | 15.7%(7.6&-23.8%) | 61.5%(34.5%−88.5%) | 0.001 |

OS: overall survival

performance status, and chemotherapy regimen (Table 6). In the univariate analysis, clinical M stage, ECOG performance status, and chemotherapy regimen were found to be statistically associated with overall survival (Fig 2)

A multivariate Cox regression model was constructed, incorporating all of the above factors. However, age, sex, tumor location, clinical M stage, and ECOG performance did not show statistically significant differences.

## Discussion

Multiple squamous cell carcinomas often arise in the upper aerodigestive tract [20,21]. The management and clinical course of patients with synchronous head and neck squamous cell carcinoma and esophageal cancer are poorly documented. Shinoto M [16] investigated an oral anticancer regimen that combines tegafur, a metabolically activated prodrug of 5-fluorouracil, with 5-chloro-2,4-dihydroxypyridine, and potassium oxonate. This regimen was found to be feasible, with a low mortality rate and acceptable morbidity for patients with synchronous Head and neck squamous cell carcinoma (HNSCC) and esophageal squamous cell carcinoma (ESCC). This provides us with insight, given the similar tissue types of esophageal cancer and head and neck cancer [22–25]. The oral anticancer regimen might be beneficial for esophageal cancer patients.

In 2016, J.-C. Lin [17] designed the novel weekly CDFLME induction chemotherapy for patients with locally advanced HNSCC, which demonstrated a higher response rate and lower incidence of Grade 3/4 mucositis and neutropenia compared to the TPF/PF regimens. In our study, this is the first clinical study to evaluate the CDFLME regimen for advanced esophageal squamous cell carcinoma (ESCC).

**Table 6. Univariate and multivariate analyses of overall survival between two groups.**

| Variable | Univariate Hazard Ratio (95% CI) | p-value | Multivariate Hazard Ratio (95% CI) | *p*-value |
|---|---|---|---|---|
| Age | 0.99(0.97-1.03) | 0.792 | 1.01(0.98-1.05) | 0.385 |
| Sex | | | | |
| Male | 1 | | 1 | |
| Female | 0.49(0.12-2.04) | 0.333 | 0.85(0.20-3.67) | 0.829 |
| Location | | | | |
| Ce | 1 | | 1 | |
| Ut | 0.83(0.19-3.64) | 0.805 | 0.34(0.07-1.68) | 0.185 |
| Mt | 1.20(0.29-5.06) | 0.802 | 0.55(0.12-2.59) | 0.446 |
| Lt | 1.11(0.26-4.64) | 0.89 | 0.43(0.09-2.11) | 0.298 |
| TNM Clinical stage | | | | |
| T | | | | |
| T1/T2 | 1 | | 1 | |
| T3/T4 | 1.78(0.96-3.32) | 0.068 | 1.63(0.55-4.81) | 0.375 |
| N | | | | |
| N- | 1 | | 1 | |
| N+ | 1.52(0.70-3.32) | 0.292 | 1.25(0.47-3.35) | 0.658 |
| M | | | | |
| M0 | 1 | | | |
| M1 | 1.88(0.18-2.99) | 0.008* | | |
| Stage | | | | |
| I | 1 | | 1 | |
| II | 0.55(0.12-2.48) | 0.439 | 0.18(0.03-1.02) | 0.052 |
| III | 1.22(0.35-4.18) | 0.756 | 0.25(0.04-1.81) | 0.171 |
| IVA | 1.88(0.55-6.42) | 0.316 | 0.37(0.05-2.75) | 0.332 |
| IVB | 2.33(0.71-7.63) | 0.161 | 0.45(0.07-3.06) | 0.417 |
| TRAE Grade | | | | |
| 1–2 | 1 | | | |
| 3–4 | 0.99(0.53-1.84) | 0.969 | | |
| ECOG status | | | | |
| **0–2** | **1** | | **1** | |
| 3–4 | 4.98(1.54-16.16) | 0.007* | 4.27(0.90-20.24) | 0.067 |
| Regimen | | | | |
| PF | 1 | | 1 | |
| CD-FL-ME | 0.25(0.10-0.63) | 0.003* | 0.22(0.08-0.64) | 0.006* |

Ce: cervical esophagus

Ut: upper thoracic esophagus

Mt: middle thoracic esophagus

Lt: lower thoracic esophagus

N-: refers to clinical stage of N0

N+: refers to clinical stages N1, N2, and N3

TRAE: treatment-related adverse event

GECOG: Eastern Cooperative Oncology Group Performance Status

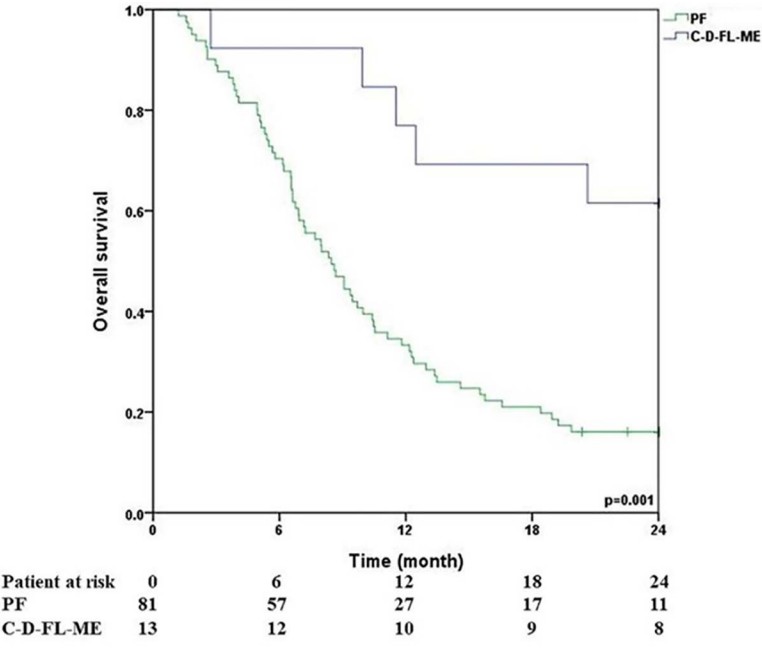

**Fig 2. Overall survival in the PF and CDFLME groups.**

The concept of dose density involves administering drugs with shortened intervals between treatments. This approach is supported by experimental models that indicate a specific dose targets a fraction of exponentially growing neoplastic cells rather than a fixed number. It is hypothesized that more frequent administration of cytotoxic therapy could be more effective in reducing residual tumor burden compared to dose escalation [26]. Based on these models, which have demonstrated survival benefits in patients with breast carcinoma [27]. We hypothesize that a dose dense regimen incorporating six active agents could be effective in improving response rates for patients with advanced esophagus cancer.

Due to the rarity of using such multidrug regimens and HNSCC cancer regimens in esophageal cancer patients, we have cautiously and on a small scale applied this approach to esophageal cancer patients within our institution. We carefully evaluate both the efficacy and toxicity of this treatment. Based on our study, this regimen demonstrated promising feasibility with a high disease control rate and improved overall survival time in advanced ESCC.

Docetaxel has shown extensive cytotoxic and antitumor activity against various common cancers in previous animal models. K. Muro [28,29] evaluated the activity and toxicity of docetaxel in metastatic esophageal cancer patients, finding it effective with careful management of neutropenia. The JCOG0807 [30,31] phase I/II trial assessed the combination of 2-weekly docetaxel with cisplatin plus fluorouracil in metastatic esophageal cancer, demonstrating promising activity and tolerability, particularly with no incidence of febrile neutropenia.

The FOLFOX regimen (oxaliplatin, leucovorin, and fluorouracil) has been widely used to treat metastatic colorectal cancer for decades. Previous studies have indicated that the FOLFOX regimen may also benefit patients with unresectable advanced esophageal squamous cell carcinoma (ESCC). A.M. Mauer [32] evaluated the efficacy and tolerability of oxaliplatin, fluorouracil, and leucovorin in patients with advanced esophageal cancer, demonstrating significant antitumor activity and a favorable toxicity profile in those with metastatic esophageal carcinoma. Thierry Conroy [15] assessed the efficacy and safety of the FOLFOX regimen compared to fluorouracil and cisplatin as part of chemoradiotherapy in patients with localized esophageal cancer. While chemoradiotherapy with FOLFOX did not increase progression free

survival compared to fluorouracil and cisplatin, FOLFOX might offer a more convenient treatment option for patients with localized esophageal cancer who are not suitable candidates for surgery.

Considering safety, the high incidence of febrile neutropenia poses a significant challenge in the 5 week PF regimen chemotherapy course, with reported rates of up to 12% [8]. Our research uncovered an intriguing observation regarding the CDFLME group. Among the 8 patients experiencing treatment-related adverse events in this group, 3 suffered only from grade 1–2 mucositis or nausea/vomiting. However, the remaining 5 patients encountered severe grade 3–4 adverse events, with 4 experiencing neutropenia and 1 facing grade 3–4 mucositis alone. Notably, neutropenia complicated by septic shock was the most common cause of treatment-related death, accounting for 1 patient (7.7%) in the CDFLME group. This underscores the heightened mortality risk associated with grade 3–4 neutropenia in this regimen. Despite these observations, the CDFLME regimen was generally well tolerated, with no significant differences observed in treatment-related deaths or grade 3–4 adverse events compared to the PF group. The notable distinction was a significant difference in grade 1–2 mucositis be-tween the two groups. Among all the mortality in the CDFLME group, only Case 5 was deemed a treatment-related death, occurring within three months of using the CDFLME regimen. Case 3, on the other hand, succumbed after undergoing chemotherapy based on the NCCN guideline's palliative regimen. The other deaths were attributed to cachexia, pneumonia, and tumor hemorrhage. Given that only one of the deaths was treatment-related and that the majority of patients passed away due to subsequent tumor progression or other factors, with an mean survival time of 19.18 months, our analysis indicates that the CDFLME regimen is safe.

In both univariate and multivariate analyses, age. sex and tumor location did not significantly affect overall survival rates. While more advanced clinical stages tended to correlate with decreased overall survival, this trend did not reach statistical significance, possibly due to the study's small sample size. Additionally, our study revealed that higher performance status scores were associated with shorter overall survival in univariate analysis, reflecting reduced chemotherapy tolerance in patients with poorer general health. However, this correlation did not remain significant in multivariate analysis, consistent with findings from previous studies. In the multivariate analysis, the results confirmed that the CDFLME regimen was beneficial for overall survival. This finding is consistent with our previous results for 1-year and 2-year OS. However, other factors, such as clinical stage, ECOG performance status, age, sex and tumor location, did not reach statistical significance, which may be due to the small sample size and selection bias.

Our study was limited to Taiwanese patients with esophageal cancer, predominantly squamous cell carcinoma of the thoracic esophagus. Patients with previous head and neck cancer who underwent definitive chemoradiotherapy were also included. There were several limitations in our study. Due to the lack of quality-of-life metrics, we were unable to comprehensively assess patients' responses after treatment. Due to the fact that some patients continued their follow-up at branch hospitals of Changhua Christian Hospital, tracking quality-of-life metrics was challenging in terms of feasibility, leading to their exclusion.

One limitation is that it was a retrospective study utilizing data from our hospital's database, which inherently introduces complexities, variations, and biases. Second, there may have been incomplete or inaccurate clinical data collection during the study period. In addition, since physicians may prefer to administer the CDFLME regimen to patients with better performance status, we observed a higher proportion of patients with better performance status (ECOG 0–2) in the CDFLME group. This may have influenced the improved survival rates and outcomes. Additionally, considering this is an alternative regimen without prior studies evaluating its efficacy and safety. We will use this regimen on a small scale and carefully select patients, which may introduce selection bias. We acknowledge that our study primarily focuses on survival and response rates and does not include quality-of-life metrics, long-term toxicity and cumulative side effects. The small sample size, the ratio of case and control groups, and the unmatched groups for confounding factors contribute to a considerable selection bias, which is a limitation of our study. In future studies, we plan to adopt randomized controlled designs to minimize selection bias as much as possible and larger, more representative cohorts thereby enhancing the credibility and reliability of the results.

## Conclusion.

In summary, our retrospective study demonstrated the efficacy of CDFLME as an alternative definitive chemotherapy for advanced ESCC. It showed promising activity and tolerability in metastatic esophageal cancer, with superior mean overall survival time, complete response rate, and disease control rate compared to the PF group. Furthermore, there were no significant differences in treatment-related deaths between the two groups, with the only notable difference being in grade 1–2 mucositis. Therefore, the CDFLME regimen could be considered an alternative treatment option for advanced ESCC. However, due to potential selection biases and the expanding landscape of treatment options that may improve efficacy and safety, further studies with larger cohorts are warranted to confirm these findings

## Author contributions

**Conceptualization:** Bing-Yen Wang.

**Data curation:** Shao-Syuan Tong, Ya-Fu Cheng, Jin-Ching Lin, Ching-Yuan Cheng, Chang-Lun Huang, Wei-Heng Hu.

**Methodology:** Yi-Ling Chen.

**Software:** Yi-Ling Chen.

**Validation:** Yi-Ling Chen.

**Writing – original draft:** Sung-Chi Yu.

**Writing – review & editing:** Bing-Yen Wang.

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
