## [Decision Letter · Decision Letter 0]

PONE-D-24-37030Efficacy and safety of cisplatin + docetaxel + 5-FU + leucovorin + methotrexate and epirubicin combination chemotherapy for advanced esophageal cancerPLOS ONE

Dear Dr. Wang,

Thank you for submitting your manuscript to PLOS ONE. After careful consideration, we feel that it has merit but does not fully meet PLOS ONE’s publication criteria as it currently stands. Therefore, we invite you to submit a revised version of the manuscript that addresses the points raised during the review process.

We look forward to receiving your revised manuscript.

Kind regards,

Jie Yang, M.D.

Guest Editor

PLOS ONE

Journal Requirements:

2. Thank you for stating the following in your Competing Interests section: [NO authors have competing interests]. Please complete your Competing Interests on the online submission form to state any Competing Interests. If you have no competing interests, please state "The authors have declared that no competing interests exist.", as detailed online in our guide for authors at http://journals.plos.org/plosone/s/submit-now This information should be included in your cover letter; we will change the online submission form on your behalf.

Reviewers' comments:

Reviewer's Responses to Questions

**Comments to the Author**

1. Is the manuscript technically sound, and do the data support the conclusions?

Reviewer #1: Partly

Reviewer #2: Yes

Reviewer #3: Partly

2. Has the statistical analysis been performed appropriately and rigorously? 

Reviewer #1: I Don't Know

Reviewer #2: Yes

Reviewer #3: Yes

3. Have the authors made all data underlying the findings in their manuscript fully available?

Reviewer #1: No

Reviewer #2: Yes

Reviewer #3: Yes

4. Is the manuscript presented in an intelligible fashion and written in standard English?

Reviewer #1: No

Reviewer #2: Yes

Reviewer #3: Yes

5. Review Comments to the Author

Reviewer #1: Dear Authors,

This is an interesting and novel research, and I appreciated the opportunity to review it. I commend your valuable effort to advance the scientific understanding of this significant health concern. Below, I have outlined my suggestions for enhancing this manuscript.

1. The ratio of case and control groups, as well as not-matched groups for confounding factors, result in a considerable selection bias of the study that highly affects the power of the study.

2. Although I believe that language should not be a barrier to knowledge transfer, I would ask the authors to improve the writing of the paper.

3. Please include the number of controls in the abstract as well.

4. In a part of the introduction section, CDFLEM is inconsistent with this in other parts of the manuscript (CDFLME). Additionally, please define it when first used.

5. “Pathology” is the correct term for examining tissue biopsies, including histological analysis. Please substitute it with histology.

6. Please outline the target organs for metastasis that you considered concerning progressive disease.

7. Please rearrange and separate the paragraphs in the introduction section based on the subject change.

8. In some parts of the manuscript, including keywords and the body of the text, “regime” is mentioned instead of regimen.

9. Please define SCCHN when first used. Furthermore, this is inconsistent with other manuscript parts (HNSCC).

10. Please provide a reference for categorizing response levels and other definitions in the methods section.

11. Please provide a reference for the treatment schedule.

12. Please clarify whether informed consent was obtained, as the current statement is unclear.

13. Please clarify that all monitoring visits were completed before the start of the study and that the data was obtained from the hospital.

14. The sentence “The results of the multivariate analysis confirmed that the CDFLME regimen was beneficial to overall survival” is an interpretation of the statistical analysis results, which is better stated in the discussion section.

15. “to our knowledge” at the beginning of the discussion section is inappropriate for the sentence.

16. Please place figures in the results section.

Best regards.

Reviewer #2: This study explores the efficacy and safety of the following combined regimen (cisplatin + docetaxel + 5-FU + leucovorin + methotrexate + epirubicin) on advanced esophageal cancer.

Study seems clear, detailed, and organized; however, please address the following comments

1. When authors stated (In current research, there are few studies that explore the efficacy and

safety of this type of head and neck cancer chemotherapeutic regimens in

advanced esophageal cancer patients). Can you please cite these studies?.

2. Can you confirm all grammars and typos' issues?.

Reviewer #3: 

1. This is a retrospective study with a relatively small sample size (94 patients, with only 13 in the CDFLME group), which limits the statistical power and generalizability. A prospective, randomized controlled trial with a larger cohort would provide more robust evidence.

2. The CDFLME group had a higher proportion of patients with better performance status (ECOG 0-2), which could have influenced the improved survival rates and outcomes. Addressing potential selection bias would improve the study's credibility.

3. While the study compares the novel CDFLME regimen to the standard of regimen, it doesn’t consider other potentially relevant combinations, such as those including immune checkpoint inhibitors, which have shown efficacy in advanced esophageal cancer.

4. The study focuses on survival and response rates but does not evaluate the patients' quality of life. Including metrics on quality of life would provide a more comprehensive view of the treatment’s impact.

5. Although toxicity is briefly discussed, more in-depth analysis and detailed reporting (such as long-term toxicity and cumulative side effects) would be valuable to assess the regimen’s safety profile.

6. A future study should ideally be a randomized controlled trial with a larger and more diverse patient population to validate these findings and minimize bias.

7. Including other treatment regimens, especially those with newer therapeutic options, would provide a more comprehensive comparison and reflect current standards in oncology.

8. Tracking the patients’ quality of life and long-term adverse effects would offer insights into the tolerability of the regimen beyond the survival outcomes.

9. Mucositis was significantly more common in the CDFLME group; additional information on supportive care protocols and preventive measures for mucositis would be helpful.

10. The data availability statement restricts access due to institutional regulations. Providing a more accessible data-sharing plan, even with de-identified data, would enhance transparency and allow for independent verification of results.

6. PLOS authors have the option to publish the peer review history of their article (what does this mean? ). If published, this will include your full peer review and any attached files.

**Do you want your identity to be public for this peer review?** For information about this choice, including consent withdrawal, please see our Privacy Policy .

Reviewer #1: **Yes: ** Sepideh Hajivalizadeh

Reviewer #2: No

Reviewer #3: No

---

## [Author Response · Author response to Decision Letter 1]

1 Dec 2024

Dear Editor and Reviewers

Thank you for spending your valuable time to review our paper. Fortunately, this paper could be submitted to PLOS one. We were grateful to Editors and Reviewers to review this paper and make some useful recommendation. That made this paper more valuable. It was our honor to have an opportunity to publish in PLOS one. We have considered the review’s comment and replied

them as below.

Review Comments to the Author

Reviewer #1: Dear Authors,

This is an interesting and novel research, and I appreciated the opportunity to review it. I commend your valuable effort to advance the scientific understanding of this significant health concern. Below, I have outlined my suggestions for enhancing this manuscript.

1. The ratio of case and control groups, as well as not-matched groups for confounding factors, result in a considerable selection bias of the study that highly affects the power of the study.

Thanks for your suggestion. Since the CDFLME regimen was applied to esophageal cancer for the first time, we conducted the study on a small scale and attempted to minimize differences between the two groups. However, the issue of a limited sample size and selection bias could not be fully addressed. In the future, we plan to include a larger number of patients to enhance the validity and statistical power of the study.

2. Although I believe that language should not be a barrier to knowledge transfer, I would ask the authors to improve the writing of the paper.

Thanks for your suggestion. We will make every effort to improve the clarity and quality of the writing in the manuscript.

3. Please include the number of controls in the abstract as well.

Thanks for your suggestion. We will include the number of controls in the abstract.

4. In a part of the introduction section, CDFLEM is inconsistent with this in other parts of the manuscript (CDFLME). Additionally, please define it when first used.

Thanks for your suggestion. We will correct this error and provide a definition when the term is first introduced in the manuscript.

5. “Pathology” is the correct term for examining tissue biopsies, including histological analysis. Please substitute it with histology.

Thanks for your suggestion. We will correct this error and make every effort to improve the quality of the writing in the manuscript.

6. Please outline the target organs for metastasis that you considered concerning progressive disease.

Thanks for your suggestion. When the tumor exhibits distant metastasis, including the spread of cancer cells to other organs such as the lungs, liver, bones, or brain, it is considered disease progression. We will provide additional clarification in the Methods section.

7. Please rearrange and separate the paragraphs in the introduction section based on the subject change.

Thanks for your suggestion, we will rearrange and separate the paragraphs in the introduction section based on the subject change.

8. In some parts of the manuscript, including keywords and the body of the text, “regime” is mentioned instead of regimen.

Thanks for your suggestion. We will correct this error and make every effort to improve the quality of the writing in the manuscript.

9. Please define SCCHN when first used. Furthermore, this is inconsistent with other manuscript parts (HNSCC).

Thanks for your suggestion. We will correct this error and standardize the terminology to HNSCC throughout the manuscript

10. Please provide a reference for categorizing response levels and other definitions in the methods section.

Thanks for your suggestion. We have cited RECIST 1.1 in the Methods section as the reference for evaluating response levels.

11. Please provide a reference for the treatment schedule.

Thanks for your suggestion. This treatment schedule was based on Dr Lin's weekly CDFLME induction chemotherapy for patients with locally advanced HNSCC. Given the favorable performance of cisplatin-based regimens in esophageal cancer patients, we adapted this regimen and applied it on a small scale to esophageal cancer patients. We will include this reference in the Discussion section and cite it as [17]

12. Please clarify whether informed consent was obtained, as the current statement is unclear.

In this retrospective study, the requirement for informed consent was waived by the Institutional Review Board (IRB) of Changhua Christian Hospital (IRB number: 231101). This waiver was granted because the study involved de-identified data collected from medical records and posed minimal risk to the participants. The approval for the study, including the waiver of informed consent, was obtained on November 21, 2023.

13. Please clarify that all monitoring visits were completed before the start of the study and that the data was obtained from the hospital.

Thanks for your suggestion. All monitoring visits were completed prior to the commencement of this retrospective study. The data used in this study were obtained from the medical records of Changhua Christian Hospital, ensuring accuracy and completeness.

14. The sentence “The results of the multivariate analysis confirmed that the CDFLME regimen was beneficial to overall survival” is an interpretation of the statistical analysis results, which is better stated in the discussion section.

Thank you for pointing this out. We will revise the manuscript by moving the sentence, "The results of the multivariate analysis confirmed that the CDFLME regimen was beneficial to overall survival," to the Discussion section, where it will be better aligned with the interpretation of the statistical analysis results.

15. “to our knowledge” at the beginning of the discussion section is inappropriate for the sentence.

Thank you for pointing this out. We will revise the sentence by removing "to our knowledge" and rephrasing it to ensure clarity and appropriateness in the context of the discussion section.

16. Please place figures in the results section.

Thank you for your suggestion. We will revise the manuscript by relocating the figures to the Results section to ensure proper alignment with the presentation of data.

Reviewer #2:

1. When authors stated (In current research, there are few studies that explore the efficacy and safety of this type of head and neck cancer chemotherapeutic regimens in advanced esophageal cancer patients). Can you please cite these studies?

Thank you for your comment. We acknowledge this limitation and will revise the manuscript to include relevant references that support our statement.

If there are no directly related studies available, we will clarify this point and provide context to highlight the gap in current research. We will cite these studies as references 16, 17, and 18.

2. Can you confirm all grammars and typos' issues?.

Thank you for your suggestion. We will thoroughly review the manuscript to ensure that all grammatical errors and typographical issues are corrected

Reviewer #3:

1. This is a retrospective study with a relatively small sample size (94 patients, with only 13 in the CDFLME group), which limits the statistical power and generalizability. A prospective, randomized controlled trial with a larger cohort would provide more robust evidence.

Thank you for your feedback. We acknowledge the limitation of the small sample size in this retrospective study, particularly the small number of patients in the CDFLME group. We have addressed this limitation in the manuscript. In the future, we plan to conduct a prospective, randomized controlled trial with a larger cohort to provide more robust and generalizable evidence.

2. The CDFLME group had a higher proportion of patients with better performance status (ECOG 0-2), which could have influenced the improved survival rates and outcomes. Addressing potential selection bias would improve the study's credibility.

Thank you for highlighting this point. We acknowledge that the higher proportion of patients with better performance status (ECOG 0-2) in the CDFLME group could have influenced the improved survival rates and outcomes. We have addressed this potential selection bias as a limitation in the manuscript and discussed its impact on the study's findings. In future studies, we aim to minimize selection bias by using randomized controlled designs to improve the credibility and reliability of the results.

3. While the study compares the novel CDFLME regimen to the standard of regimen, it doesn’t consider other potentially relevant combinations, such as those including immune checkpoint inhibitors, which have shown efficacy in advanced esophageal cancer.

Thank you for your valuable comment. In our study, both groups were treated according to the NCCN guidelines. For patients meeting the criteria for immunotherapy, immune checkpoint inhibitors were administered regardless of whether they received neoadjuvant chemoradiotherapy (NCRT). This approach ensures consistency in treatment decisions and avoids bias introduced by treatment selection. We will clarify this point in the manuscript.

4. The study focuses on survival and response rates but does not evaluate the patients' quality of life. Including metrics on quality of life would provide a more comprehensive view of the treatment’s impact.

Thank you for your insightful comment. We acknowledge that our study primarily focuses on survival and response rates and does not include quality-of-life metrics. This is a limitation of the current study, and we will address it in the manuscript. In future research, we plan to incorporate validated quality-of-life assessment tools to provide a more comprehensive evaluation of the treatment's impact on patients.

5. Although toxicity is briefly discussed, more in-depth analysis and detailed reporting (such as long-term toxicity and cumulative side effects) would be valuable to assess the regimen’s safety profile.

Thank you for your valuable suggestion. While our current study provides a brief discussion on toxicity, we acknowledge the importance of a more in-depth analysis and detailed reporting, including long-term toxicity and cumulative side effects, to comprehensively assess the safety profile of the regimen. We will revise the discussion section to address this limitation and include it as a recommendation for future research

6. A future study should ideally be a randomized controlled trial with a larger and more diverse patient population to validate these findings and minimize bias.

Thank you for your valuable suggestion. We completely agree that a future study should ideally be a randomized controlled trial with a larger and more diverse patient population to validate our findings and minimize bias. We have acknowledged this point as a limitation in our current study and plan to design future research accordingly.

7. Including other treatment regimens, especially those with newer therapeutic options, would provide a more comprehensive comparison and reflect current standards in oncology.

Thanks for your suggestion. We agree that those newer therapy options will provide valuable reference for our future research.

8. Tracking the patients’ quality of life and long-term adverse effects would offer insights into the tolerability of the regimen beyond the survival outcomes.

Thank you for your valuable suggestion. we acknowledge the importance of patients’ quality of life and long-term adverse effects to comprehensively assess the safety profile of the regimen. We will revise the discussion section to address this limitation and include it as a recommendation for future research

9. Mucositis was significantly more common in the CDFLME group; additional information on supportive care protocols and preventive measures for mucositis would be helpful.

Thank you for your suggestion. Mucositis is indeed a significant concern in the CDFLME group. We will provide additional details on the supportive care protocols and preventive measures for managing mucositis in our next update. Your feedback is valuable and will help us ensure comprehensive coverage of this important aspect.

10. The data availability statement restricts access due to institutional regulations. Providing a more accessible data-sharing plan, even with de-identified data, would enhance transparency and allow for independent verification of results.

Thank you for your feedback. We understand the importance of data accessibility for transparency and independent verification. While institutional regulations currently limit direct data sharing, we will explore options to provide de-identified data or alternative access methods within the allowed framework. Your suggestion is greatly appreciated and will guide us in improving future data-sharing plans.

---

## [Decision Letter · Decision Letter 1]

PONE-D-24-37030R1Efficacy and safety of cisplatin + docetaxel + 5-FU + leucovorin + methotrexate and epirubicin combination chemotherapy for advanced esophageal cancerPLOS ONE

Dear Dr. Wang,

Thank you for submitting your manuscript to PLOS ONE. After careful consideration, we feel that it has merit but does not fully meet PLOS ONE’s publication criteria as it currently stands. Therefore, we invite you to submit a revised version of the manuscript that addresses the points raised during the review process.

We look forward to receiving your revised manuscript.

Kind regards,

Jie Yang, M.D.

Guest Editor

PLOS ONE

Additional Editor Comments:

Please further address the reviewers' concerns.

Reviewers' comments:

Reviewer's Responses to Questions

**Comments to the Author**

1. If the authors have adequately addressed your comments raised in a previous round of review and you feel that this manuscript is now acceptable for publication, you may indicate that here to bypass the “Comments to the Author” section, enter your conflict of interest statement in the “Confidential to Editor” section, and submit your "Accept" recommendation.

Reviewer #1: (No Response)

Reviewer #2: All comments have been addressed

Reviewer #3: All comments have been addressed

2. Is the manuscript technically sound, and do the data support the conclusions?

Reviewer #1: Yes

Reviewer #2: Yes

Reviewer #3: Yes

3. Has the statistical analysis been performed appropriately and rigorously? 

Reviewer #1: I Don't Know

Reviewer #2: Yes

Reviewer #3: Yes

4. Have the authors made all data underlying the findings in their manuscript fully available?

Reviewer #1: No

Reviewer #2: Yes

Reviewer #3: Yes

5. Is the manuscript presented in an intelligible fashion and written in standard English?

Reviewer #1: Yes

Reviewer #2: Yes

Reviewer #3: Yes

6. Review Comments to the Author

Reviewer #1: Dear authors,

I appreciate your addressing my previous comments. Nevertheless, the manuscript could still benefit from some revision. Below, I have provided my comments regarding its improvement.

1. Please define CDFLME when first used and remove the definition in the other part of the Introduction and Methods.

2. Please specify the exact sites you considered as “tumor progression” in the part you added to the methods section regarding tumor metastasis. If you have stated all of them, please substitute "including" for "such as."

3. It is better to mention the date for the statement “J.-C. Lin designed a novel weekly CDFLME induction chemotherapy regimen” in the Introduction (e.g., In 2016, J.-C. Lin designed…)

4. The definition of “partial response” in the RECIST guideline version 1.1 is as follows: At least a 30% decrease in the sum of diameters of target lesions, taking as reference the baseline sum diameters. Please revise it.

5. Please provide a reference for the definitions of “treatment-related death” and “the disease control rate.”

6. Please provide references for treatment schedules administered to both groups of patients in the Methods section.

7. Please mention the explanation you stated in the “Author's Response To Reviewer Comments” section regarding informed consent in the Methods section as well (In this retrospective study, the requirement for informed consent was waived by the Institutional Review Board (IRB) of Changhua Christian Hospital (IRB number: 231101)).

8. Please provide the statement, “All monitoring visits were completed prior to the commencement of this retrospective study. The data used in this study were obtained from the medical records of Changhua Christian Hospital, ensuring accuracy and completeness” in the Methods section.

9. I couldn’t find “The results of the multivariate analysis confirmed that the CDFLME regimen was beneficial to overall survival” in the Discussion section.

10. Please place both “Fig. 1” and “Fig. 2” where they are referred to in the text.

11. In the Discussion section, please discuss all the variables you measured in your study and mentioned in the Results section.

12. Please mention the small sample size, the ratio of case and control groups, as well as not-matched groups for confounding factors, which result in a considerable selection bias, as limitations of your study.

Best regards.

Reviewer #2: (No Response)

Reviewer #3: The author has responded thoroughly to reviewer comments, below is a general assessment and areas that still need improvement in the manuscript. After this reccomended for the publication.

General Comments:

1. While the authors acknowledge and explain the limitations of a small sample size and selection bias, the manuscript could further elaborate on how these issues might be mitigated in future studies.

2. Despite assurances of language improvement, some sentences remain verbose or awkwardly phrased. Additional proofreading is necessary for clarity and grammatical correctness.

3. Although the authors agreed to include control numbers in the abstract, the revised text does not clearly specify this detail. Ensure this is updated.

4. Moving figures to the results section as suggested is critical for better alignment with data presentation. Confirm that this change has been made appropriately.

5. While the authors addressed inconsistencies like "CDFLEM" vs. "CDFLME" and "SCCHN" vs. "HNSCC," double-check that all terms are now consistent throughout the manuscript.

6. The authors have clarified many points, but it is advisable to further detail why the chosen chemotherapy regimen is innovative compared to others, such as immune checkpoint inhibitors.

7. Adding a brief acknowledgment of the lack of quality-of-life metrics and why they were excluded (e.g., feasibility or retrospective nature) would strengthen the discussion.

8. The manuscript includes a restrictive data availability statement, which could be improved by providing clearer alternatives for data access.

7. PLOS authors have the option to publish the peer review history of their article (what does this mean? ). If published, this will include your full peer review and any attached files.

**Do you want your identity to be public for this peer review?** For information about this choice, including consent withdrawal, please see our Privacy Policy .

Reviewer #1: **Yes: ** Sepideh Hajivalizadeh

Reviewer #2: No

Reviewer #3: No

---

## [Author Response · Author response to Decision Letter 2]

19 Feb 2025

Review Comments to the Author

Reviewer #1: Dear authors,

I appreciate your addressing my previous comments. Nevertheless, the manuscript could still benefit from some revision. Below, I have provided my comments regarding its improvement.

1. Please define CDFLME when first used and remove the definition in the other part of the Introduction and Methods.

Thank you for your feedback. I have defined CDFLME at its first mention and removed the redundant definitions from the Introduction and Methods sections.

2. Please specify the exact sites you considered as “tumor progression” in the part you added to the methods section regarding tumor metastasis. If you have stated all of them, please substitute "including" for "such as."

Thank you for your suggestion. I have specified the exact sites considered as 'tumor progression' in the Methods section. I have replaced 'such as' with 'including' as recommended.

3. It is better to mention the date for the statement “J.-C. Lin designed a novel weekly CDFLME induction chemotherapy regimen” in the Introduction (e.g., In 2016, J.-C. Lin designed…)

Thank you for your suggestion. I have added the relevant date to the statement in the Introduction as recommended (e.g., 'In 2016, J.-C. Lin designed a novel weekly CDFLME induction chemotherapy regimen').

4. The definition of “partial response” in the RECIST guideline version 1.1 is as follows: At least a 30% decrease in the sum of diameters of target lesions, taking as reference the baseline sum diameters. Please revise it.

Thank you for your suggestion. I have revised the sentence for clarity as follows: ' partial response is defined as at least a 30% decrease in the sum of diameters of target lesions.'

5. Please provide a reference for the definitions of “treatment-related death” and “the disease control rate.”

Thank you for your suggestion. Since there is no consensus on a temporal definition of treatment-related death , we adopted the stricter criteria of Adelstein et al. (Adelstein DJ, Li Y, Adams GL, et al. An intergroup phase III comparison of standard radiation therapy and two schedules of concurrent chemoradiotherapy in patients with unresectable squamous cell head and neck cancer. J Clin Oncol 2003; 21: 92–98.) defining TRD as death occurring from CRT start until a month from the end of CRT.

Disease Control Rate (DCR): The proportion of patients who achieve a response to treatment, including complete response (CR), partial response (PR), and stable disease (SD), according to RECIST criteria

6. Please provide references for treatment schedules administered to both groups of patients in the Methods section.

Thank you for your suggestion. In the Methods section, the treatment schedule for the PF regimen has been referenced from [3], while the CDFLME regimen has been referenced from [17]

7. Please mention the explanation you stated in the “Author's Response To Reviewer Comments” section regarding informed consent in the Methods section as well (In this retrospective study, the requirement for informed consent was waived by the Institutional Review Board (IRB) of Changhua Christian Hospital (IRB number: 231101)).

Thank you for your suggestion. I have incorporated the explanation regarding informed consent into the Methods section as follows: 'In this retrospective study, the requirement for informed consent was waived by the Institutional Review Board (IRB) of Changhua Christian Hospital (IRB number: 231101)

8. Please provide the statement, “All monitoring visits were completed prior to the commencement of this retrospective study. The data used in this study were obtained from the medical records of Changhua Christian Hospital, ensuring accuracy and completeness” in the Methods section.

Thank you for your suggestion. I have added the following statement to the Methods section as recommended: 'All monitoring visits were completed prior to the commencement of this retrospective study. The data used in this study were obtained from the medical records of Changhua Christian Hospital, ensuring accuracy and completeness.

9. I couldn’t find “The results of the multivariate analysis confirmed that the CDFLME regimen was beneficial to overall survival” in the Discussion section.

Thank you for your feedback. I have added the following statement to the Discussion section: 'In the multivariate analysis, the results confirmed that the CDFLME regimen was beneficial for overall survival. This finding is consistent with our previous results for 1-year and 2-year OS. However, other factors, such as clinical stage, ECOG performance status, age, and tumor location, did not reach statistical significance, which may be due to the small sample size and selection bias

10. Please place both “Fig. 1” and “Fig. 2” where they are referred to in the text.

Thank you for your suggestion. I have adjusted the placement of 'Fig. 1' and 'Fig. 2' to align with their first mentions in the text.

11. In the Discussion section, please discuss all the variables you measured in your study and mentioned in the Results section.

Thank you for your suggestion. I have revised the Discussion section to ensure that all variables measured in the study and mentioned in the Results section are appropriately discussed.

12. Please mention the small sample size, the ratio of case and control groups, as well as not-matched groups for confounding factors, which result in a considerable selection bias, as limitations of your study

Thank you for your suggestion. I have added the following sentence to the Discussion section: 'The small sample size, the ratio of case and control groups, and the unmatched groups for confounding factors contribute to a considerable selection bias, which is a limitation of our study.

Reviewer #3: The author has responded thoroughly to reviewer comments, below is a general assessment and areas that still need improvement in the manuscript. After this reccomended for the publication.

General Comments:

1. While the authors acknowledge and explain the limitations of a small sample size and selection bias, the manuscript could further elaborate on how these issues might be mitigated in future studies.

Thank you for your suggestion. I have expanded the Discussion section to further elaborate on how the issues of small sample size and selection bias can be mitigated in future studies, including the potential use of randomized controlled designs and larger, more representative cohorts.

2. Despite assurances of language improvement, some sentences remain verbose or awkwardly phrased. Additional proofreading is necessary for clarity and grammatical correctness.

Thank you for your feedback. I will conduct additional proofreading to improve clarity and ensure grammatical correctness, refining any verbose or awkwardly phrased sentences

3. Although the authors agreed to include control numbers in the abstract, the revised text does not clearly specify this detail. Ensure this is updated.

Thank you for your suggestion. I have updated the abstract to clearly specify the control numbers as follows: ' Among them, 81 patients received fluorouracil + cisplatin regimen serving as the control group.' Please let me know if any further modifications are needed.

4. Moving figures to the results section as suggested is critical for better alignment with data presentation. Confirm that this change has been made appropriately.

Thank you for your suggestion. I have moved the figures to the Results section to better align with data presentation as recommended. Please let me know if any further adjustments are needed.

5. While the authors addressed inconsistencies like "CDFLEM" vs. "CDFLME" and "SCCHN" vs. "HNSCC," double-check that all terms are now consistent throughout the manuscript.

Thank you for your careful review. I have double-checked the manuscript to ensure that all terms, including 'CDFLME' and 'HNSCC,' are now consistently used throughout. Please let me know if any further modifications are needed.

6. The authors have clarified many points, but it is advisable to further detail why the chosen chemotherapy regimen is innovative compared to others, such as immune checkpoint inhibitors.

Thank you for your suggestion. The innovation of our chemotherapy regimen lies in its original use for head and neck cancer patients. In 2016, J.C. Lin demonstrated its efficacy in this population, which led to its small-scale application in esophageal cancer patients.

7. Adding a brief acknowledgment of the lack of quality-of-life metrics and why they were excluded (e.g., feasibility or retrospective nature) would strengthen the discussion.

Thank you for your suggestion. I have added a brief acknowledgment in the Discussion section as follows: Due to the fact that some patients continued their follow-up at branch hospitals of Changhua Christian Hospital, tracking quality-of-life metrics was challenging in terms of feasibility, leading to their exclusion.

8. The manuscript includes a restrictive data availability statement, which could be improved by providing clearer alternatives for data access.

Thank you for your feedback. In the future, we will communicate with the institution and explore clearer alternative solutions for data access to improve this aspect

---

## [Decision Letter · Decision Letter 2]

PONE-D-24-37030R2Efficacy and safety of cisplatin + docetaxel + 5-FU + leucovorin + methotrexate and epirubicin combination chemotherapy for advanced esophageal cancerPLOS ONE

Dear Dr. Wang,

Thank you for submitting your manuscript to PLOS ONE. After careful consideration, we feel that it has merit but does not fully meet PLOS ONE’s publication criteria as it currently stands. Therefore, we invite you to submit a revised version of the manuscript that addresses the points raised during the review process.

We look forward to receiving your revised manuscript.

Kind regards,

Jie Yang, M.D.

Guest Editor

PLOS ONE

Journal Requirements:

Reviewers' comments:

Reviewer's Responses to Questions

**Comments to the Author**

1. If the authors have adequately addressed your comments raised in a previous round of review and you feel that this manuscript is now acceptable for publication, you may indicate that here to bypass the “Comments to the Author” section, enter your conflict of interest statement in the “Confidential to Editor” section, and submit your "Accept" recommendation.

Reviewer #1: (No Response)

Reviewer #3: All comments have been addressed

2. Is the manuscript technically sound, and do the data support the conclusions?

Reviewer #1: Yes

Reviewer #3: Yes

3. Has the statistical analysis been performed appropriately and rigorously? 

Reviewer #1: I Don't Know

Reviewer #3: Yes

4. Have the authors made all data underlying the findings in their manuscript fully available?

Reviewer #1: No

Reviewer #3: Yes

5. Is the manuscript presented in an intelligible fashion and written in standard English?

Reviewer #1: Yes

Reviewer #3: Yes

6. Review Comments to the Author

Reviewer #1: Dear authors,

I appreciate your addressing my previous comments. Nevertheless, the manuscript could still benefit from some revision. Below, I have provided my comments regarding its improvement.

1. CDFLME is still not defined correctly. Please notice that the Abstract and the body of the text are two different parts. Hence, CDFLME should be defined once in the Abstract and once in the body of the text, both when first used. Please consider this writing policy precisely.

2. Regarding my previous comment, [“Please specify the exact sites you considered as “tumor progression” in the part you added to the methods section regarding tumor metastasis. If you have stated all of them, please substitute "including" for "such as."], still “including” is not substituted for “such as.” Please consider this carefully.

3. Please add “taking as reference the baseline sum diameters” to the end of the statement “at least a 30% decrease in the sum of diameters of target lesions.”

4. Unfortunately, I was unable to find a precise treatment schedule administered to the PF group in reference [3], as you mentioned. Furthermore, the treatment schedule described in reference differs from the one administered to the CDFLME group in your study in terms of the number of days.

5. Please remove the appended “in the Methods section” from the end of “All monitoring visits were completed prior to the commencement of this retrospective study. The data used in this study were obtained from the medical records of Changhua Christian Hospital, ensuring accuracy and completeness”.

Best regards.

Reviewer #3: (No Response)

7. PLOS authors have the option to publish the peer review history of their article (what does this mean? ). If published, this will include your full peer review and any attached files.

**Do you want your identity to be public for this peer review?** For information about this choice, including consent withdrawal, please see our Privacy Policy .

Reviewer #1: **Yes: ** Sepideh Hajivalizadeh

Reviewer #3: No

---

## [Author Response · Author response to Decision Letter 3]

27 Mar 2025

Dear Editor and Reviewers

Thank you for spending your valuable time to review our paper. Fortunately, this paper could be submitted to PLOS one. We were grateful to Editors and Reviewers to review this paper and make some useful recommendation. That made this paper more valuable. It was our honor to have an opportunity to publish in PLOS one. We have considered the review’s comment and replied

them as below.

Reviewer #1: Dear authors,

I appreciate your addressing my previous comments. Nevertheless, the manuscript could still benefit from some revision. Below, I have provided my comments regarding its improvement.

1. CDFLME is still not defined correctly. Please notice that the Abstract and the body of the text are two different parts. Hence, CDFLME should be defined once in the Abstract and once in the body of the text, both when first used. Please consider this writing policy precisely.

Thank you for your feedback. I have revised the manuscript accordingly by defining CDFLME both in the Abstract and again at its first appearance in the main body of the text, as per the writing policy. Please let me know if any further adjustments are needed.

2. Regarding my previous comment, [“Please specify the exact sites you considered as “tumor progression” in the part you added to the methods section regarding tumor metastasis. If you have stated all of them, please substitute "including" for "such as."], still “including” is not substituted for “such as.” Please consider this carefully.

Thank you for your valuable comment. In accordance with your suggestion, we have replaced “such as” with “including” and clearly specified the sites considered indicative of tumor progression.

We have revised the sentence as follows: "When a tumor exhibits distant metastasis to sites including the lungs, liver, bones, distal lymph nodes, adrenal gland, or brain, it is considered disease progression."

3. Please add “taking as reference the baseline sum diameters” to the end of the statement “at least a 30% decrease in the sum of diameters of target lesions.”

Thank you for your suggestion. We have revised the sentence accordingly by adding “taking as reference the baseline sum diameters” to the end of the statement “at least a 30% decrease in the sum of diameters of target lesions,” as recommended.

4 .Unfortunately, I was unable to find a precise treatment schedule administered to the PF group in reference [3], as you mentioned. Furthermore, the treatment schedule described in reference differs from the one administered to the CDFLME group in your study in terms of the number of days.

Thank you for your comment. We acknowledge that the treatment schedule for the PF group was not clearly described in reference [3], as you pointed out. Additionally, we recognize that the schedule differed from that of the CDFLME group in terms of the number of treatment days.

To address this, we have replaced reference [3] with the NCCN Clinical Practice Guidelines in Oncology (NCCN Guidelines™) Esophageal Cancers and Esophagogastric Junction (excluding the proximal 5 cm of the stomach), Version 3.2023, which provides a more reliable and standardized reference.

Moreover, we have revised the description of the treatment schedule for the CDFLME group to clarify the differences in treatment days.

5.please remove the appended “in the Methods section” from the end of “All monitoring visits were completed prior to the commencement of this retrospective study. The data used in this study were obtained from the medical records of Changhua Christian Hospital, ensuring accuracy and completeness”.

Thank you for your suggestion. We have removed the appended phrase “in the Methods section” from the end of the sentence as requested.

---

## [Decision Letter · Decision Letter 3]

PONE-D-24-37030R3Efficacy and safety of cisplatin + docetaxel + 5-FU + leucovorin + methotrexate and epirubicin combination chemotherapy for advanced esophageal cancerPLOS ONE

Dear Dr. Wang,

Thank you for submitting your manuscript to PLOS ONE. After careful consideration, we feel that it has merit but does not fully meet PLOS ONE’s publication criteria as it currently stands. Therefore, we invite you to submit a revised version of the manuscript that addresses the points raised during the review process.

We look forward to receiving your revised manuscript.

Kind regards,

Jie Yang, M.D.

Guest Editor

PLOS ONE

**Journal Requirements:**

Reviewers' comments:

Reviewer's Responses to Questions

**Comments to the Author**

1. If the authors have adequately addressed your comments raised in a previous round of review and you feel that this manuscript is now acceptable for publication, you may indicate that here to bypass the “Comments to the Author” section, enter your conflict of interest statement in the “Confidential to Editor” section, and submit your "Accept" recommendation.

Reviewer #1: (No Response)

2. Is the manuscript technically sound, and do the data support the conclusions?

Reviewer #1: Yes

3. Has the statistical analysis been performed appropriately and rigorously? 

Reviewer #1: I Don't Know

4. Have the authors made all data underlying the findings in their manuscript fully available?

Reviewer #1: No

5. Is the manuscript presented in an intelligible fashion and written in standard English?

Reviewer #1: Yes

6. Review Comments to the Author

**Reviewer #1: ** Dear authors,

I appreciate your addressing my previous comments. Nevertheless, the manuscript could still benefit from some revision. Below, I have provided my comments regarding its improvement.

1. Please clarify whether you administered the revised version of the treatment schedule for the CDFLME group or your previous schedule. Please note that it should be stated exactly as it was administered during the study. Furthermore, the new schedule is not consistent with Figure 1.

2. Can you please provide the PDF of the NCCN Clinical Practice Guidelines in Oncology (NCCN GuidelinesTM) Esophageal Cancers and Esophagogastric Junction (Excluding the proximal 5cm of the stomach) Version 3.2023? I was unable to get access to it.

Best regards.

7. PLOS authors have the option to publish the peer review history of their article (what does this mean? ). If published, this will include your full peer review and any attached files.

**Do you want your identity to be public for this peer review?** For information about this choice, including consent withdrawal, please see our Privacy Policy .

Reviewer #1: **Yes: ** Sepideh Hajivalizadeh

---

## [Author Response · Author response to Decision Letter 4]

4 May 2025

Reviewer #1: Dear authors,

I appreciate your addressing my previous comments. Nevertheless, the manuscript could still benefit from some revision. Below, I have provided my comments regarding its improvement.

1. Please clarify whether you administered the revised version of the treatment schedule for the CDFLME group or your previous schedule. Please note that it should be stated exactly as it was administered during the study. Furthermore, the new schedule is not consistent with Figure 1.

We appreciate the reviewer’s observation. We confirm that the treatment schedule administered to the CDFLME group was the original version, as described in the Methods section of the manuscript. We apologize for the confusion caused by the discrepancy between the revised schedule and Figure 1. To maintain consistency and accuracy, we have corrected Figure 1 to reflect the actual treatment protocol used during the study. All related descriptions have been revised accordingly in the revised manuscript.

2. Can you please provide the PDF of the NCCN Clinical Practice Guidelines in Oncology (NCCN GuidelinesTM) Esophageal Cancers and Esophagogastric Junction (Excluding the proximal 5cm of the stomach) Version 3.2023? I was unable to get access to it.

We sincerely thank the reviewer for pointing this out. Upon careful review, we acknowledge that we mistakenly cited the NCCN Clinical Practice Guidelines in Oncology (Version 3.2023) in our manuscript. In fact, the version we used and referenced throughout the study was Version 2.2023. We have corrected all citations accordingly in the revised manuscript to accurately reflect this. We apologize for the oversight and appreciate your attention to this detail.

The accurate citation is:

Ajani JA, D'Amico TA, Almhanna K, et al. NCCN Clinical Practice Guidelines in Oncology: Esophageal and Esophagogastric Junction Cancers, Version 2.2023. J Natl Compr Canc Netw. 2023 Apr;21(4):393–422. doi: 10.6004/jnccn.2023.0019.

We apologize for the oversight and appreciate your attention to this detail.

---

## [Decision Letter · Decision Letter 4]

Efficacy and safety of cisplatin + docetaxel + 5-FU + leucovorin + methotrexate and epirubicin combination chemotherapy for advanced esophageal cancer

PONE-D-24-37030R4

Dear Dr. Wang,

We’re pleased to inform you that your manuscript has been judged scientifically suitable for publication and will be formally accepted for publication once it meets all outstanding technical requirements.

Kind regards,

Jie Yang, M.D.

Guest Editor

PLOS ONE

Additional Editor Comments (optional):

Thanks for the authors' efforts to comprehensively improve your manuscript according to editor's and reviewers' comments. I am pleased to inform you that your paper can be accepted for publication now. Thanks for the chance to assess your interesting and important work. Additionally, many thanks for all the reviewers' precious inputs.

Reviewers' comments:

Reviewer's Responses to Questions

**Comments to the Author**

1. If the authors have adequately addressed your comments raised in a previous round of review and you feel that this manuscript is now acceptable for publication, you may indicate that here to bypass the “Comments to the Author” section, enter your conflict of interest statement in the “Confidential to Editor” section, and submit your "Accept" recommendation.

Reviewer #1: All comments have been addressed

2. Is the manuscript technically sound, and do the data support the conclusions?

Reviewer #1: Yes

3. Has the statistical analysis been performed appropriately and rigorously? 

Reviewer #1: I Don't Know

4. Have the authors made all data underlying the findings in their manuscript fully available?

Reviewer #1: No

5. Is the manuscript presented in an intelligible fashion and written in standard English?

Reviewer #1: Yes

6. Review Comments to the Author

Reviewer #1: (No Response)

7. PLOS authors have the option to publish the peer review history of their article (what does this mean? ). If published, this will include your full peer review and any attached files.

**Do you want your identity to be public for this peer review?** For information about this choice, including consent withdrawal, please see our Privacy Policy .

Reviewer #1: **Yes: ** Sepideh Hajivalizadeh

---

## [Editor Report · Acceptance letter]

PONE-D-24-37030R4

PLOS ONE

Dear Dr. Wang,

I'm pleased to inform you that your manuscript has been deemed suitable for publication in PLOS ONE. Congratulations! Your manuscript is now being handed over to our production team.

Kind regards,

on behalf of

Dr. Jie Yang

Guest Editor

PLOS ONE